# Evaluating Creative Language Generation: The Case of Rap Lyric Ghostwriting

## Abstract

Language generation tasks that seek to mimic human ability to use language creatively are difficult to evaluate, since one must consider creativity, style, and other non-trivial aspects of the generated text. The goal of this paper is to develop evaluation methods for one such task, ghostwriting of rap lyrics, and to provide an explicit, quantifiable foundation for the goals and future directions of this task. Ghostwriting must produce text that is similar in style to the emulated artist, yet distinct in content. We develop a novel evaluation methodology that addresses several complementary aspects of this task, and illustrate how such evaluation can be used to meaningfully analyze system performance. We provide a corpus of lyrics for 13 rap artists, annotated for stylistic similarity, which allows us to assess the feasibility of manual evaluation for generated verse.

## 1 Introduction

Language generation tasks are often among the most difficult to evaluate. Evaluating machine translation, image captioning, summarization, and other similar tasks is typically done via comparison with existing human-generated "references". However, human beings also use language creatively, and for the language generation tasks that seek to mimic this ability, determining how accurately the generated text represents its target is insufficient, as one also needs to evaluate creativity and style. We believe that one of the reasons such tasks receive little attention is the lack of sound evaluation methodology, without which no task is well-defined, and no progress can be made. The goal of this paper is to develop an evaluation methodology for one such task, ghostwriting, or more specifically, ghostwriting of rap lyrics.

Ghostwriting is ubiquitous in politics, literature, and music. As such, it introduces a distinction between the performer/presenter of text, lyrics, etc, and the creator of text/lyrics. The goal of ghostwriting is to present something in a style that is believable enough to be credited to the performer. In the domain of rap specifically, rappers sometimes function as ghostwriters early on before embarking on their own public careers, and there are even businesses that provide written lyrics as a service[1]. The goal of automatic ghostwriting is therefore to create a system that can take as input a given artist's work and generate **similar** yet **unique** lyrics.

Our objective in this work is to provide a quantifiable direction and foundation for the task of rap lyric generation and similar tasks through (1) developing an evaluation methodology for such models, and (2) illustrating how such evaluation can be used to analyze system performance, including advantages and limitations of a specific language model developed for this task. As an illustration case, we use the ghostwriter model previously proposed in exploratory work by Potash et al. (2015), which uses a recurrent neural network (RNN) with Long Short-Term Memory (LSTM) for rap lyric generation.

The following are the main contributions of this paper. We present a comprehensive manual evaluation methodology of the generated verses along three key aspects: fluency, coherence, and style matching. We introduce an improvement to the semi-automatic methodology used by Potash et al. (2015) that automatically penalizes repetitive text, which removes the need for manual intervention

---

[1] http://www.rap-rebirth.com/, http://www.precisionwrittens.com/rap-ghostwriters-for-hire/

and enables a large-scale analysis. Finally, we build a corpus of lyrics for 13 rap artists, each with his own unique style, and conduct a comprehensive evaluation of the LSTM model performance using the new evaluation methodology. The corpus includes style matching annotation for select verses in dataset, which can form a gold standard for future work on automatic representation of similarity between artists' styles. The resulting rap lyric dataset is publicly available from the authors' website.

Additionally, we believe that the annotation method we propose for manual style evaluation can be used for other similar generation tasks. One example is 'Deep Art' work in the computer vision community that seeks to apply the style of a particular painting to other images (Gatys et al., 2015; Li and Wand, 2016). One of the drawbacks of such work is a lack of systematic evaluation. For example, Li and Wand (2016) compared the results of the model with previous work by doing a manual inspection during an informal user study. The presence of a systematic formal evaluation process would lead to a clearer comparison between models and facilitate progress in this area of research. With this in mind, we make the interface used for style evaluation in this work available for public use.

Our evaluation results highlight the truly multifaceted nature of the ghostwriting task. While having a single measure of success is clearly desirable, our analysis shows the need for complementary metrics that evaluate different components of the overall task. Indeed, despite the fact that our test-case LSTM model outperforms a baseline model across numerous artists based on automated evaluation, the full set of evaluation metrics is able to showcase the LSTM model's strengths and weaknesses. The coherence evaluation demonstrates the difficulty of incorporating large amounts of training data into the LSTM model, which intuitively would be desirable to create a flexible ghostwriting model. The style matching experiments suggest that the LSTM is effective at capturing an artist's general style. However, this may indicate that it tends to form 'average' verses, which are then more likely to be matched with existing verses from an artist rather than another random verse from the same artist. Overall, the evaluation methodology we present provides an explicit, quantifiable foundation for the ghostwriting task, allowing for a deeper understanding of the task's goals and future research directions.

## 2 Related Work

In the past few years there has been a significant amount of work dedicated to the evaluation of natural language generation (Hastie and Belz, 2014), dealing with different aspects of evaluation methodology. However, most of this work focuses on simple tasks, such as referring expressions generation. For example, Belz and Kow (2011) investigated the impact of continuous and discrete scales for generated weather descriptions, as well as and simple image descriptions that typically consist of a few words (e.g., `the small blue fan`). Perplexity is arguably the most popular automated metric for language models. Unfortunately, work such as Vougiouklis et al. (2016) shows that perplexity can sometimes be negatively correlated with human evaluation.

Previous work that explores text generation for artistic purposes, such as poetry and lyrics, generally uses either automated or manual evaluation. In terms of manual evaluation, Barbieri et al. (2012) have a set of annotators evaluate generated lyrics along two separate dimensions: grammar and semantic relatedness to song title. The annotators rated the dimensions with scores 1-3. A similar strategy was used by Gervás (2000), where the author had annotators evaluate generated verses with regard to syntactic correctness and overall aesthetic value, providing scores in the range 1-5. Wu et al. (2013) had annotators determine the effectiveness of various systems based on fluency as well as rhyming.

Some heuristic-based automated approaches have also been used. For example, Oliveira et al. (2014) use a simple automatic heuristic that awards lines for ending in a termination previously used in the generated stanza. Malmi et al. (2015) evaluate their generated lyrics based on the verses' rhyme density, on the assumption that a higher rhyme density means better lyrics.

Note that none of the work cited above provide a comprehensive evaluation methodology, but rather focus on certain specific aspects of generated verses, such as rhyme density or syntactic correctness. Moreover, the methodology for generating lyrics, proposed by the various authors, in-

fluences the evaluation process. For instance, Barbieri et al. (2012) did not evaluate the presence of rhymes because the model was constrained to produce only rhyming verses. Furthermore, none of the aforementioned works implement models that generate complete verses at the token level (including verse structure), which is the goal of the models we aim to evaluate. In contrast to previous approaches that evaluate whole verses, our evaluation methodology uses a fine-grained, line-by-line scheme, which makes it easier for human annotators, as they no longer need to evaluate the whole verse at once. In addition, despite the fact the each line is annotated using a discrete scale, our methodology produces a continuous numeric score for the whole verse, enabling better comparison.

## 3   Dataset

For our evaluation experiments, we selected the following list of artists in four different categories: three top-selling rap artists(Eminem, Jay Z, Tupac), artists with the largest (Aesop Rock, GZA, Sage Francis) and the smallest (DMX, Drake) vocabulary, and best classified artists from Hirjee and Brown (2010b) (Fabolous, Nototious B.I.G., Lil' Wayne).

We collected all available songs from the above artists from the site *The Original Hip-Hop (Rap) Lyrics Archive - OHHLA.com - Hip-Hop Since 1992*[2]. We removed the metadata, line repetiton markup, and chorus lines, and tokenized the lyrics using the NLTK library (Bird et al., 2009). Since the preprocessing was done heuristically, the resulting dataset may still contain some text that is not actual verse, but rather dialogue or chorus lines. We therefore filter out all verses that are shorter than 20 tokens. Statistics of our dataset are shown in Table 1. We have made the dataset publicly available[3].

## 4   Evaluation Methodology

We believe that adequate evaluation for the ghost-writing task requires both manual and automatic approaches. The automated evaluation methodology enables large-scale analysis of the generated verse. However, given the nature of the task, the automated evaluation is not able to assess certain critical aspects of fluency and style, such as the

---

[2] http://www.ohhla.com/
[3] Left blank for review.

vocabulary, tone, and themes preferred by a particular artist. In this section, we present a manual methodology for evaluating these aspects of the generated verse, as well as an improvement to the automatic methodology proposed by Potash et al. (2015).

### 4.1   Manual Evaluation

We have designed two annotation tasks for manual evaluation. The first task is to determine how fluent and coherent the generated verses are. The second task is to evaluate manually how well the generated verses match the style of the target artist.

**Fluency/Coherence Evaluation**   Given a generated verse, we ask annotators to determine the fluency and coherence of the lyrics. Even though our evaluation is for systems that produce entire verses, we follow the work of Wu (2014) and annotate fluency, as well as coherence, at the line level. To assess fluency, we ask to what extent a given line can be considered a valid English utterance. Since a language model may produce highly disjointed verses as it progresses through the training process, we offer the annotator three options for grading fluency: strongly fluent, weakly fluent, and not fluent. If a line is disjointed, i.e., it is only fluent in specific segments of the line, the annotators are instructed to mark it as weakly fluent. The grade of not fluent is reserved for highly unintelligible text.

To assess coherence, we ask the annotator how well a given line matches the preceding line. That is, how believable is it that these two lines would follow each other in a rap verse. We offer the annotators the same choices as in the fluency evaluation: strongly coherent, weakly coherent, and not coherent. During the training process, a language model may output the same line repeatedly. We account for this in our coherence evaluation by defining the consecutive repetition of a line as not coherent. This is important to define because the line on its own may be strongly fluent, however, a coherent verse cannot consist of a single fluent line repeated indefinitely.

**Style Matching**   The goal of the style matching annotation is to determine how well a given verse captures the style of the target artist. In this annotation task, a user is presented with an evaluation verse and asked to compare it against four other verses. The goal is to pick the verse that is written in a similar style. One of the four choices

| Artist | Verses | Unique Vocab | Vocab Richness | Avg Len | Stdev Len | Max Len |
|---|---|---|---|---|---|---|
| Tupac | 660 | 5776 | 7.1 | 117 | 83 | 423 |
| Aesop Rock | 549 | 11815 | 14.8 | 140 | 139 | 1039 |
| DMX | 819 | 5593 | 5.3 | 125 | 82 | 552 |
| Drake | 665 | 6064 | 7.0 | 128 | 112 | 1057 |
| Eminem | 1429 | 12393 | 6.2 | 136 | 105 | 931 |
| Fabolous | 892 | 8304 | 7.4 | 122 | 91 | 662 |
| GZA | 287 | 6845 | 15.9 | 145 | 102 | 586 |
| Jay Z | 1245 | 9596 | 6.7 | 111 | 81 | 842 |
| Lil' Wayne | 1564 | 10848 | 5.5 | 124 | 101 | 977 |
| Nototious B.I.G. | 426 | 5465 | 10.2 | 120 | 88 | 557 |
| Sage Francis | 570 | 8082 | 11.9 | 114 | 112 | 645 |
| Kanye West | 851 | 7007 | 7.6 | 105 | 109 | 2264 |
| Kool Keith | 1471 | 13280 | 7.4 | 118 | 85 | 626 |
| Too Short | 1259 | 7396 | 4.3 | 134 | 123 | 1411 |

Table 1: Rap lyrics dataset statistics. Vocabulary richness measures how varied an artist's vocabulary is, computed as the total number of words divided by vocabulary size.

is always a verse from the same artist that was used to generate the verse being evaluated. The other three verses are chosen from the remaining artists in our dataset. Each verse is evaluated in this manner four times, each time against different verses, so that it has the chance to get matched with a verse from each of the remaining twelve artists. The generated verse is considered stylistically consistent if the annotators tend to select the verse that belongs to the target artist. To evaluate the difficulty of this task, we also perform style matching annotation for authentic verse, in which the evaluated verse is not generated, but rather is an actual existing verse from the target artist. [4]

### 4.2 Automated Evaluation

The automated evaluation we describe below attempts to capture computationally the dual aspects of "unique yet similar" in a manner originally proposed by Potash et al. (2015).

**Uniqueness of Generated Lyrics** We use a modified tf-idf representation for verses, and calculate cosine similarity between generated verses and the verses from the training data to determine novelty (or lack thereof). In order to more directly penalize generated verses that are primarily the reproduction of a single verse from the training set, we calculate the maximum similarity score across all training verses. That is, we do not want generated verses that contain text from a single training verse, which in turn rewards generated verses that draw from numerous training verses.

**Stylistic Similarity via Rhyme Density of Lyrics**
We use the rhyme density method proposed by Hirjee and Brown (2010a) to evaluate how well the generated verse models an artist's style. The point of an effective system is not to produce arbitrary rhymes: it is to produce rhyme types and rhyme frequency similar to the target artist. Distilling the various rhyme metrics offered by the tool, we focus on rhyme density, which is defined as the number of rhymed syllables divided by the total number of syllables (Hirjee and Brown, 2010a). Certain artists distinguish themselves by having more complicated rhyme schemes, such as the use of internal[5] or polysyllabic rhymes[6]. Rhyme density is able to capture this in a single metric, since the tool we use is able to detect these various forms of rhymes. Moreover, as was shown in Wu (2014), the inter-annotator agreement (IAA) for manual rhyme detection is low (the highest IAA was only 0.283 using a two-scale annotation scheme), which is expected due to the subjective nature of the task. Therefore, an objective automatic methodology is desirable. Since this tool is trained on a distinct corpus of lyrics, it can provide a "uniform" experience and give an impartial and objective score.

However, the rhyme detection tool is not designed to deal with highly repetitive text, which the LSTM model produces often in the early stages of training. Since the same phoneme is repeated (because the same word is repeated), the

---

[4]We have made the annotation interface available on (https://github.com/placeholder).

[5]e.g. "New York City gritty committee pity the fool" and "How I made it you salivated over my calibrated"

[6]e.g. "But it was your op to shop stolen art/Catch a swollen heart form not rolling smart".

rhyme detection tool generates a false positive. Potash et al. (2015) deal with this by manually inspecting the rhyme densities of verses generated in the early stages of training to determine if a generated verse should be kept for the evaluation procedure. Unfortunately, This approach is clearly not scalable, as it is not fully automatic.

In order to fully automate this method, we propose to handle highly repetitive text by weighting the rhyme density of a given verse by its entropy. More specifically, for a given verse, we calculate entropy at the token level and divide by total number of tokens in that verse. Verses with highly repetitive text will have a low entropy, which results in down-weighting the rhyme density of verses that produce false positive rhymes due to their repetitive text.

**Merging Uniqueness and Similarity** Since ghostwriting is a balancing act of the two opposing forces of textual uniqueness and stylistic similarity, we want a low correlation between rhyme density (stylistic similarity) and maximum verse similarity (lack of textual uniqueness). However, our goal is not to have a high rhyme density, but rather to have a rhyme density similar to the target artist, while simultaneously keeping the maximum similarity score low. As the model overfits the training data, both the value of maximum similarity and the rhyme density will increase, until the model generates the original verse directly. Therefore, our goal is to evaluate the value of the maximum similarity at the point where the rhyme density has the value of the target artist. In order to accomplish this, we follow Potash et al. (2015) and plot the values of rhyme density and maximum similarity obtained at different points during model training. We use regression lines for these points to identify the value of the maximum similarity line at the point where the rhyme density line has the value of the target artist. We give more detail below.

## 5 Lyric Generation Experiments

The main generative model we use in our evaluation experiments is an LSTM. Similar to Potash et al. (2015), we use an n-gram model as a baseline system for automated evaluation. We refer the reader to the original work for a detailed description. After every 100 iterations of training[7] the

---

[7]Training is done in batches with two verses per iteration. Therefore, since each artist has a different training set, the number of iterations that constitutes a full epoch also varies.

LSTM model generates a verse. For the baseline model, we generate five verses at values 1-9 for $n$. We see a correspondence between higher $n$ and higher iteration: as both increase, the models become more 'fit' to the training data.

For the baseline model, we use the verses generated at different n-gram lengths ($n \in \{1, ..., 9\}$) to obtain the values for regression. At every value of $n$, we take the average rhyme density and maximum similarity score of the five verses that we generate to create a single data point for rhyme density and maximum similarity score, respectively.

To enable comparison, we also create nine data points from the verses generated by the LSTM as follows. A separate model for each artist is trained for a minimum of 16,400 iterations. We take the verses generated every 2,000 iterations, from 0 to 16,000 iterations, giving us nine points. The averages for each point are obtained by using the verses generated in iterations $\pm x, x \in \{100, 200, 300, 400\}$ for each interval of 2,000.

## 6 Results

### 6.1 Fluency/Coherence

In order to fairly compare the fluency/coherence of verses across artists, we use the verses generated by each artist's model at 16,000 iterations. We apply the fluency/coherence annotation methodology from Section 4.1. Each line is annotated by two annotators. Annotation results are shown in Figure 1 and Figure 2. For each annotated verse, we report the percentage of lines annotated as strongly fluent, weakly fluent, and not fluent, as well as the corresponding percentages for coherence. We convert the raw annotation results into a single score for each verse by treating the labels "strongly fluent", "weakly fluent", and "not fluent" as numeric values 1, 0.5, and 0, respectively. Treating each annotation on a given line separately, we calculate the average numeric rating for a given verse: $Fluency = \frac{\#sf + 0.5\#wf}{\#a}$ where $\#sf$ is the number of times any line is labeled strongly fluent, $\#wf$ is the number of times any line is labeled weakly fluent, and $\#a$ is the total annotations provided for a verse, which is equal to the number of lines $\times$ 2. *Coherence* is calculated in a similar manner. Raw inter-annotator agreement (IAA) for fluency annotation was 0.67. For coherence annotation, the IAA was 0.43. We believe coherence has a lower agreement because it is more

semantic, as opposed to syntactic, in nature, causing it to be more subjective. Note that while the agreement is relatively low, it is expected, given the subjective nature of the task. For example, Wu (2014) report similar agreement values for the fluency annotation they perform.

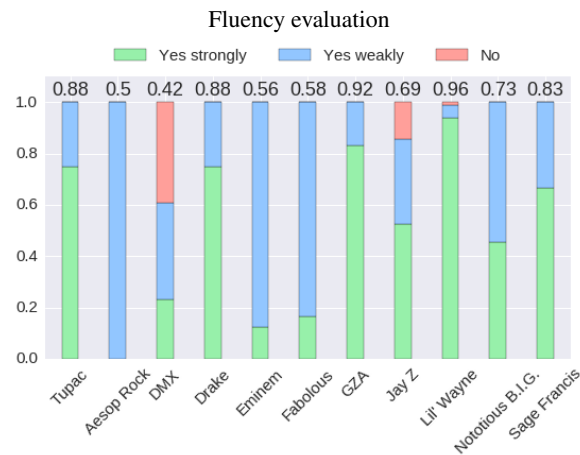

Figure 1: Percentage of lines annotated as strongly fluent, weakly fluent, and not fluent. The numbers above the bars reflect the total score of the artist (higher is better). The resulting mean is 0.723 and the standard deviation is 0.178.

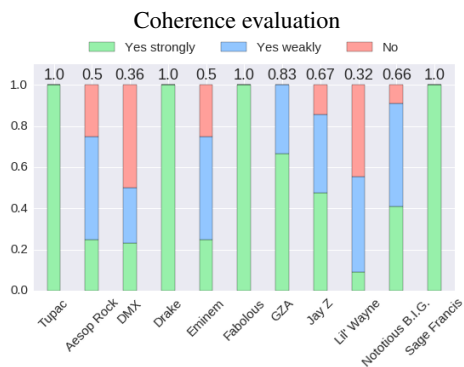

Figure 2: Percentage of lines annotated as strongly coherent, weakly coherent, and not coherent. The numbers above the bars reflect the total score of the artist (higher is better). The resulting mean is 0.713 and the standard deviation is 0.256.

## 6.2 Style Matching

We performed style-matching annotation for the verses generated at iterations 16,000–16,400 for each artist. For the experiment with authentic verses, we randomly chose five verses from each artist, with a verse length of at least 40 tokens. Each page was annotated twice, by native English-speaking rap fans. The results of our style matching annotations are shown in Table 2. We present two different views of the results. First, each annotation for a page is considered separately and

we calculate: $Match\% = \frac{\#m}{\#a}$, where $\#m$ is the number of times, on a given page, the chosen verse actually came from the target artist, and $\#a$ is the total number of annotations done. For a given artist, five verses were evaluated, each verse appeared on four separate pages, and each page is annotated twice, so $\#a$ is equal to 40. Since in each case (i.e., page) the classes are different, we cannot use Fleiss's kappa directly. Raw agreement for style annotation, which corresponds to the percentage of times annotators picked the same verse (whether or not they are correct) is shown in the column 'Raw agreement %' in Table 2.

We also report annotators' joint ability to guess the target artist correctly, which we compute as follows: $Match_A\% = \frac{\#m_A}{\#s_A}$, where $\#s_A$ is the number of times the annotators agreed on a verse on the same page, and $\#m_A$ is the number of times that the agreed upon verse is from the target artist.

### 6.2.1 Artist Confusion

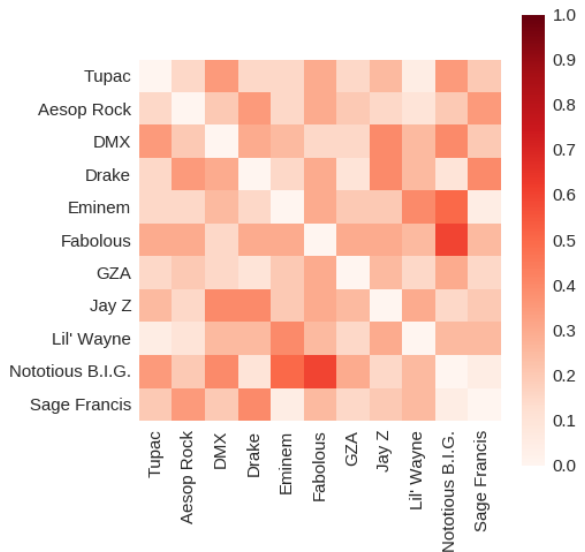

Figure 3: Fraction of confusions between artists

The results of style-matching annotation also provides us with an interesting insight into the similarity between two artists' styles. This is captured by the *confusion* between two artists during the annotation of the pages with authentic verses, which is computed as follows:

$$Confusion(a, b) = \frac{\#c(a, b) + \#c(b, a)}{\#p(a, b) + \#p(b, a)} \quad (1)$$

where $\#p(a, b)$ is the number of times a verse from artist $a$ is presented for evaluation and a verse

| Artist | Authentic | | | Generated | | |
|--------|-----------|-----------|----------------|-----------|-----------|----------------|
|        | Match% | Match$_A$% | Raw agreement % | Match% | Match$_A$% | Raw agreement % |
| Tupac | 35.0 | 50.0 | 40.0 | 45.0 | 57.1 | 35.0 |
| Aesop Rock | 30.0 | 25.0 | 40.0 | 37.5 | 100.0 | 10.0 |
| DMX | 40.0 | 71.4 | 35.0 | 27.5 | 30.0 | 50.0 |
| Drake | 32.5 | 44.4 | 45.0 | 37.5 | 40.0 | 25.0 |
| Eminem | 12.5 | 00.0 | 50.0 | 35.0 | 50.0 | 30.0 |
| Fabolous | 25.0 | 12.5 | 40.0 | 45.0 | 50.0 | 40.0 |
| GZA | 52.5 | 72.7 | 55.0 | 32.5 | 22.2 | 45.0 |
| Jay Z | 35.0 | 42.9 | 35.0 | 22.5 | 22.2 | 45.0 |
| Lil' Wayne | 27.5 | 22.2 | 45.0 | 37.5 | 57.1 | 35.0 |
| Notorious B.I.G. | 25.0 | 0.00 | 35.0 | 27.5 | 33.3 | 30.0 |
| Sage Francis | 52.5 | 66.7 | 45.0 | 22.5 | 16.7 | 30.0 |
| **Average** | 33.4 | 37.1 | 42.3 | 33.6 | 43.5 | 34.1 |

Table 2: The percentage of correct matches and the inter-annotator agreement in style matching evaluation

from artist $b$ is shown as one of four choices; $\#c(a, b)$ is the number of times the verse from artist $b$ was chosen as the matching verse. The resulting confusion matrix is presented in Figure 3. We intend for this data to provide a gold standard for future experiments that would attempt to encode the similarity of artists' styles.

### 6.3 Automated Evaluation

The results of our automated evaluation are shown in Table 3. For each artist, we calculate their average rhyme density across all verses. We then use this value to determine at which iteration this rhyme density is achieved during generation (using the regression line for rhyme density). Next, we use the maximum similarity regression line to determine the maximum similarity score at that iteration. Low maximum similarity score indicates that we have maintained stylistic similarity while producing new, previously unseen lyrics.

Note the presence of negative numbers in Table 3. The reason is that in the beginning of training (in the LSTM's case) and at a low n-gram length (for the baseline model), the models actually achieved a rhyme density that exceeded the artist's average rhyme density. As a result, the rhyme density regression line hits the average rhyme density on a negative iteration.

### 7 Discussion

In order to better understand the interaction between the four metrics we have introduced in this paper, we examined correlations between different measures of quality for generated verse (see Table 4a). The lack of strong correlation supports the notion that different aspects of verse quality should be addressed separately. Moreover, the metrics

are in fact complementary. Even the measures of *fluency* and *coherence*, despite sharing a similar goal, have a relatively low correlation of 0.4. Such low correlations emphasize our contribution, since other works (Barbieri et al., 2012; Wu, 2014; Malmi et al., 2015) do not provide a comprehensive evaluation methodology, and evaluate just one or two particular aspects. For example, Wu (2014) evaluated only fluency and rhyming, and Barbieri et al. (2012) evaluated only syntactic correctness and semantic relatedness to the title, whereas we present complementary approaches for evaluating different aspects of the generated verses.

Interestingly, the number of verses a rapper has in our dataset has a strong negative correlation with coherence score (cf. Table 4b). This can be explained by the following consideration: on iteration 16,000, the model for the authors with the smaller number of verses has seen the same verses more times than the model trained on a larger number of verses. Therefore, it is easier for the former to produce more coherent lyrics since it saw more of the same patterns. As a result, models trained on a larger number of verses have a lower coherence score. For example, Lil' Wayne has the most verses in our data, and correspondingly, the model for his verse has the worst coherence score. Note that the fluency score does not have this negative correlation with the number of verses. Based on our evaluation, 16,000 iterations is enough to learn a language model for any artist that produces fluent lines, regardless of training set size. However, these lines will not necessarily form a coherent verse if the artist has a large number of verses.

As can be seen from Table 2, the $Match\%$ score suggests that the LSTM-generated verses are able to capture the style of the artist as well as the orig-

| Artist | Avg Rhyme Density | Baseline | | LSTM | |
|---|---|---|---|---|---|
| | | Similarity | N-gram | Similarity | iteration |
| Tupac | 0.302 | **0.024** | $-2$ | 0.065 | $-3168$ |
| Aesop Rock | 0.349 | 0.745 | 7 | **0.460** | 12 470 |
| DMX | 0.341 | 0.663 | 6 | **0.431** | 8271 |
| Drake | 0.341 | 0.586 | 5 | **0.519** | 9949 |
| Eminem | 0.325 | 0.337 | 3 | **0.302** | 8855 |
| Fabolous | 0.360 | 1.353 | 14 | **0.569** | 14 972 |
| GZA | 0.280 | **0.520** | 4 | 0.616 | 14 939 |
| Jay Z | 0.365 | 0.499 | 5 | **0.463** | 15 147 |
| Lil' Wayne | 0.362 | 0.619 | 6 | **0.406** | 9249 |
| Notorious B.I.G. | 0.383 | 0.701 | 7 | **0.428** | 3723 |
| Sage Francis | 0.415 | 0.764 | 8 | **0.241** | $-187$ |
| **Average** | - | 0.619 | - | **0.409** | - |

Table 3: The results of the automated evaluation. The bold indicates the system with a lower similarity at the target rhyme density.

| | Coherence | Fluency | Similarity | Matching |
|---|---|---|---|---|
| **Coherence** | 1.000 | 0.398 | 0.102 | -0.285 |
| **Fluency** | 0.398 | 1.000 | 0.137 | -0.276 |
| **Similarity** | 0.102 | 0.137 | 1.000 | 0.092 |
| **Matching** | -0.285 | -0.276 | 0.092 | 1.000 |

(a) Correlation between the four metrics we have developed: Coherence, Fluency, similarity score based on automated evaluation (Similarity), and Style Matching (Matching).

| | Coherence | Fluency | Similarity | Matches |
|---|---|---|---|---|
| **Verses** | -0.509 | -0.084 | 0.133 | 0.111 |
| **Tokens** | -0.463 | -0.229 | -0.012 | 0.507 |
| **Vocab Richness** | 0.214 | 0.116 | -0.263 | 0.107 |

(b) Correlation between the number of verses/tokens and average coherence, fluency, and similarity scores, as well as $Match_A\%$ at 16000 iterations.

Table 4: Correlations

inal verses. Furthermore, $Match_A\%$ is significantly higher for the LSTM model, which means that the annotators agreed on matching verses more frequently. We believe this means that the LSTM model, trained on all verses from a given artist, is able to capture the artist's "average" style, whereas authentic verses represent a random selection that are less likely, statistically speaking, to be similar to another random verse. Note that, as we expect, there is a strong correlation between the number of tokens in the artist's data and the frequency of agreed-upon correct style matches (cf. Table 4b). Since verses vary in length, this correlation is not observed for verses. Finally, the lack of strong correlation with vocabulary richness suggests that token uniqueness is not as important as the sheer volume.

Lastly, we note that the fully automated methodology we propose is able to replicate the results of the previously available semi-automatic method for the rapper Fabolous, which was the

only artist evaluated by Potash et al. (2015). Furthermore, the results of automated evaluation for the 11 artists confirm that the LSTM model generalizes better than the baseline model.

## 8 Conclusion

In this paper, we have presented a comprehensive evaluation methodology for the task of ghostwriting rap lyrics, which captures complementary aspects of this task and its goals. We developed a manual evaluation method that assesses several key properties of generated verse, and created a data set of authentic verse, manually annotated for style matching. A previously proposed semi-automatic evaluation method has now been fully automated, and shown to replicate results of the original method. We have illustrated how the proposed evaluation methodology can be used to inspect an LSTM-based ghostwriter model. We believe our evaluation experiments also clearly demonstrate that complementary evaluation methods are required to capture different aspects of the ghostwriting task.

Lastly, our evaluation provides key insights into future directions for generative models. For example, the automated evaluation shows how the LSTM's inability to integrate new vocabulary makes it difficult to achieve truly desirable similarity scores; future models can draw on the work of Graves (2013) and Bowman et al. (2015) in an attempt to leverage other artists' lyrics.

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
