# Peer review of "Evaluating Creative Language Generation: The Case of Rap Lyric Ghostwriting"

_ACL 2017 — decision unknown_

[Official Review · Reviewer 1 · rating 3 · confidence 4]
soundness 5 · originality 5 · clarity 4 · impact 3 · substance 4 · appropriateness 4 · meaningful comparison 3 · presentation format Poster

This paper presents evaluation metrics for lyrics generation exploring the need
for the lyrics to be original,but in a similar style to an artist whilst being
fluent and co-herent. The paper is well written and the motivation for the
metrics are well explained.  

The authors describe both hand annotated metrics (fluency, co-herence and
match) and an automatic metric for ‘Similarity'. Whilst the metric for
Similarity is unique and interesting the paper does not give any evidence of
this as an effective automatic metric as correlations between this metric and
the others are low, (which they say that they should be used separately). The
authors claim it can be used to meaningfully analyse system performance but we
have to take their word for it as again there is no correlation with any
hand-annotated performance metric.  Getting worse scores than a baseline system
isn’t evidence that the metric captures quality (e.g. you could have a very
strong baseline).

Some missing references, e.g. recent work looking at automating co-herence,
e.g. using mutual information density (e.g. Li et al. 2015). In addition, some
reference to style matching from the NLG community are missing (e.g. Dethlefs
et al. 2014 and the style matching work by Pennebaker).

[Official Review · Reviewer 2 · rating 2 · confidence 4]
soundness 5 · originality 5 · clarity 4 · impact 3 · substance 2 · appropriateness 5 · meaningful comparison 3 · presentation format Poster

This paper studies how to properly evaluate systems that produce ghostwriting
of rap lyrics.
The authors present manual evaluation along three key aspects: fluency,
coherence, and style matching.
They also introduce automatic metrics that consider uniqueness via maximum
training similarity, and stylistic similarity via rhyme density.

I can find some interesting analysis and discussion in the paper.
The way for manually evaluating style matching especially makes sense to me.

There also exist a few important concerns for me.

I am not convinced about the appropriateness of only doing fluency/coherence
ratings at line level.
The authors mention that they are following Wu (2014), but I find that work
actually studying a different setting of hip hop lyrical challenges and
responses, which should be treated at line level in nature.
While in this work, a full verse consists of multiple lines that normally
should be topically and structurally coherent.
Currently I cannot see any reason why not to evaluate fluency/coherence for a
verse as a whole.

Also, I do not reckon that one should count so much on automatic metrics, if
the main goal is to ``generate similar yet unique lyrics''.
For uniqueness evaluation, the calculations are performed on verse level.
However, many rappers may only produce lyrics within only a few specific topics
or themes.
If a system can only extract lines from different verses, presumably we might
also get a fluent, coherent verse with low verse level similarity score, but we
can hardly claim that the system ``generalizes'' well.
For stylistic similarity with the specified artist, I do not think rhyme
density can say it all, as it is position independent and therefore may not be
enough to reflect the full information of style of an artist.

It does not seem that the automatic metrics have been verified to be well
correlated with corresponding real manual ratings on uniqueness or stylistic
matching.
I also wonder if one needs to evaluate semantic information commonly expressed
by a specified rapper as well, other than only caring about rhythm.

Meanwhile, I understand the motivation for this study is the lack of *sound*
evaluation methodology.
However, I still find one statement particularly weird:
``our methodology produces a continuous numeric score for the whole verse,
enabling better comparison.''
Is enabling comparisons really more important than making slightly vague but
more reliable, more convincing judgements?

Minor issue:
Incorrect quotation marks in Line 389

[Official Review · Reviewer 3 · rating 2 · confidence 4]
soundness 5 · originality 5 · clarity 3 · impact 3 · substance 2 · appropriateness 4 · meaningful comparison 3 · presentation format Poster

This paper proposes to present a more comprehensive evaluation methodology for
the assessment of automatically generated rap lyrics (as being similar to a
target artist).  While the assessment of the generation of creative work is
very challenging and of great interest to the community, this effort falls
short of its claims of a comprehensive solution to this problem.

All assessment of this nature ultimately falls to a subjective measure -- can
the generated sample convince an expert that the generated sample was produced
by the true artist rather than an automated preocess?  This is essentially a
more specific version of a Turing Test.   The effort to automate some parts of
the evaluation to aid in optimization and to understand how humans assess
artistic similarity is valuable.  However, the specific findings reported in
this work do not encourage a belief that these have been reliably identified.

Specifically -- Consider the central question: Was a sample generated by a
target artist?        The human annotators who were asked this were not able to
consistently respond to this question.        This means either 1) the annotators did
not have sufficient expertise to perform the task, or 2) the task was too
challenging, or some combination of the two.  

The proposed automatic measures also failed to show a reliable agreement to
human raters performing the same task.        This dramatically limits their efficacy
in providing a proxy for human assessment.   The low interannotator agreement
may be "expected" because the task is subjective, but the idea of decomposing
the evaluation into fluency and coherence components is meant to make it more
tractable, and thereby improve the consistency of rater scores.  A low IAA for
an evaluation metric is a cause for concern and limits its viability as a
general purpose tool.  

Specific questions/comments:

* Why is a line-by-line level evaluation prefered to a verse level analysis. 
Specifically for "coherence", a line by line analysis limits the scope of
coherence to consequtive lines.

* Style matching -- This term assumes that these 13 artists each have a
distinct style, and always operate in that style. I would argue that some of
these artists (kanye west, eminem, jay z, drake, tupac and notorious big) have
produced work in multiple styles.  A more accurate term for this might be
"artist matching".

* In Section 4.2 The central automated component of the evaluation is low
tf*idf with existing verses, and similar rhyme density.  Given the limitations
of rhyme density -- how well does this work.  Even with the manual intervention
described?

* In Section 6.2 -- This description should include how many judges were used
in this study? In how many cases did the judges already know the verse they
were judging?  In this case the test will not assess how easy it is to match
style, but rather, the judges recall and rap knowledge.